# Treating Pythiosis with Antibacterial Drugs Targeting Protein Synthesis: An Overview

**DOI:** 10.3390/jof10040234

**Published:** 2024-03-22

**Authors:** Erico S. Loreto, Juliana S. M. Tondolo, Régis A. Zanette

**Affiliations:** 1Sobresp Faculty of Health Sciences, 520 Appel Street, Santa Maria 97015-030, RS, Brazil; jutondolo@yahoo.com.br; 2Department of Pharmacology, Basic Health Sciences Institute, Federal University of Rio Grande do Sul, 2600 Ramiro Barcelos Street, Porto Alegre 90035-003, RS, Brazil; regnitro@yahoo.com.br

**Keywords:** *Pythium insidiosum*, *Pythiosis treatment*, protein-inhibiting antibacterials, azithromycin, linezolid

## Abstract

This review article explores the effectiveness of antibacterial drugs that inhibit protein synthesis in treating pythiosis, a difficult-to-treat infection caused by *Pythium insidiosum*. The article highlights the susceptibility of *P. insidiosum* to antibacterial drugs, such as macrolides, oxazolidinones, and tetracyclines. We examine various studies, including in vitro tests, experimental infection models, and clinical case reports. Based on our synthesis of these findings, we highlight the potential of these drugs in managing pythiosis, primarily when combined with surgical interventions. The review emphasizes the need for personalized treatment strategies and further research to establish standardized testing protocols and optimize therapeutic approaches.

## 1. Introduction

*Pythium insidiosum*, the causative agent of pythiosis, represents a significant threat to human and animal health due to its aggressive and destructive nature. This pathogen, a member of the oomycetes, is particularly concerning because of its capacity to infect and induce life-threatening conditions in otherwise healthy individuals [1,2].

Pythiosis is a significant concern for animal health, especially for horses, dogs, and, to a lesser extent, cats and other mammals [3]. In horses, the disease manifests as chronic, debilitating subcutaneous lesions, leading to systemic illness and significant tissue damage in the limbs, abdomen, and face. In dogs, it causes ulcerations, lymph node involvement, and severe gastrointestinal issues, such as tumor-like masses in the intestines, resulting in severe diarrhea, lethargy, and potentially fatal outcomes if left untreated [2].

In humans, the impact of pythiosis can be profoundly severe, necessitating procedures like enucleation for ocular infections [4] and limb amputation in cases of arteritis due to vascular commitment [1]. The disease’s severity is further heightened by its ability to spread systemically. For instance, untreated vascular pythiosis can extend through the arteries, affecting the iliac and renal arteries and even the abdominal aorta, leading to disseminated pythiosis, which is often fatal [5].

Treating *P. insidiosum* infections presents significant challenges, but growing research suggests promise for antibacterial agents that specifically target protein synthesis [6,7,8,9]. This approach could improve treatment strategies for specific forms of pythiosis. This review focuses on treating pythiosis, specifically examining the role of antibacterials that inhibit protein synthesis against *P. insidiosum*. It evaluates their effectiveness by analyzing in vitro susceptibility data, experimental infection models, and clinical case studies.

## 2. Overview of Pythiosis Treatment

The treatment of pythiosis has seen considerable advancements, yet it remains a complex challenge, primarily due to *P. insidiosum’s* resistance to traditional antifungal therapies (Table 1). Surgery is considered a key strategy, particularly effective for localized, smaller, and superficial lesions, though its success can be constrained [5]. Antifungal drugs, initially used based on the microorganism’s misclassification as a fungus, have demonstrated limited efficacy [10,11]. Immunotherapy emerges as a promising alternative, particularly beneficial in equine and some human cases, although its success varies depending on factors like lesion duration and antigen preparation methods [11,12]. Moreover, exploring adjuvant treatments and plant-derived compounds has introduced new possibilities for pythiosis management, yet their safety and effectiveness warrant further investigation [11].

## 3. *P. insidiosum* Cell Structure and Susceptibility to Antibacterial Drugs

Although *P. insidiosum* shares morphological traits with filamentous fungi, it is more closely related to organisms such as brown algae and diatoms. As a member of the Stramenopiles-Alveolata–Rhizaria supergroup, *P. insidiosum* is characterized by its broad hyphae, perpendicular branching, and the production of biflagellate zoospores in aquatic environments (Figure 1) [42,43,44].

The biochemical distinction of oomycetes from fungi is evident in its cell wall composition, which contains minimal chitin but is rich in cellulose and β-glucans [45,46,47]. Other differences include its mitochondrial structure, actin cytoskeleton, and protein repertoire [48,49,50]. Notably, *P. insidiosum* has an incomplete sterol biosynthesis pathway, relying on external sterol sources for physiological functions, which contributes to its resistance or reduced susceptibility to sterol biosynthesis inhibitors and sterol-binding drugs [21,51,52,53].

*Pythium* spp. are distinguished from true fungi due to their heightened sensitivity to antibacterials that act on protein synthesis in the 70S ribosome, such as tetracycline, chloramphenicol, streptomycin, and erythromycin [54,55,56]. It is plausible that these antimicrobials inhibit the growth of the microorganism by interfering with cytoplasmic and mitochondrial protein synthesis [55]. However, the addition of sterols (ergosterol, cholesterol, beta-sitosterol, or stigmasterol) to the culture medium shields *Pythium* spp. against the action of these antibacterials [57] and other anti-*Pythium* drugs [53], possibly by altering cell membrane permeability and reducing the entry of these drugs into the cell.

The clinical implications of these findings regarding tetracycline, chloramphenicol, streptomycin, and erythromycin are nuanced by the potential interaction between *Pythium* spp. and host-derived sterols. This protective mechanism suggests that the observed in vitro sensitivity may not directly translate to in vivo efficacy, as the possible incorporation of host-derived sterols could alter the susceptibility of *Pythium* to these drugs. However, further scientific research is required to support or refute this hypothesis.

## 4. In Vitro Anti-*Pythium* Antimicrobial Activity of Protein Synthesis-Inhibiting Antibacterials

It is crucial to recognize that, to date, a standardized susceptibility testing protocol specifically for *P. insidiosum* has yet to be established. The susceptibility assessments are adapted mainly from established protocols for fungi and bacteria. Investigations in this domain have examined diverse culture media, varying inoculum concentrations, and various methodologies [58]. These investigations have also extended to analyzing pathogenic microorganisms isolated from animal and human hosts. The heterogeneity in these testing methodologies underscores the intricate challenges and complexities that *P. insidiosum* presents in clinical microbiology and infectious disease research.

### 4.1. Anti-Pythium Antimicrobial Activity Determined by Reduction in Mycelial Weight

Marchant and Smith [54] described that chloramphenicol exerted an inhibitory effect on the growth rate of *Pythium ultimum*. The maximum inhibitory response was observed at 100 µg/mL. Rawn and Van Etten [55] investigated the sensitivity of a *P. ultimum* isolate to several antibiotics over a 24 h treatment period. They found that cycloheximide, an eukaryotic protein synthesis inhibitor, inhibited 98% of *P. ultimum* growth at a concentration of 1 µg/mL. Tetracycline showed 83% inhibition at 10 µg/mL and 99% at 100 µg/mL. Chloramphenicol resulted in 62% growth inhibition at 100 µg/mL, while erythromycin achieved 70% inhibition at 10 µg/mL and 91% at 100 µg/mL.

McMeekin [59] reported that 100 µg/mL of streptomycin could stimulate the growth of a *P. aphanidermatum* isolate, in contrast to 200 µg/mL of streptomycin, which inhibited the growth of this microorganism. Similarly, McMeekin and Mendoza [60] found varying effects of streptomycin on the in vitro growth of two *P. insidiosum* isolates, with one isolate inhibited and the other stimulated by this aminoglycoside.

### 4.2. Anti-Pythium Antimicrobial Activity Determined by Linear or Radial Growth Inhibition

Marchant and Smith [54] found that while chloramphenicol at 100 µg/mL had a lesser impact on the *P. ultimum* linear growth rate compared to its effect on dry weight production, it significantly altered the morphology, resulting in a lower density of hyphae and reduced aerial mycelium.

During the initial standardization of a disk-diffusion test to assess the susceptibility of *P. insidiosum* to antibacterials, Tondolo et al. [61] observed a unique response to minocycline (30 µg). Not only did the disks inhibit the growth of *P. insidiosum*, they also induced a phenomenon of mycelial “escape” from the antibacterial drug, as illustrated in Figure 2. Given its simplicity, the authors proposed this technique as a screening tool to distinguish *P. insidiosum* from true fungi, highlighting that true fungi exhibited no inhibition in radial growth at this minocycline concentration.

Two pivotal studies by Loreto et al. [7] and Bagga et al. [8] have provided insightful data on the antibacterial efficacy against *P. insidiosum*, assessed through the disk diffusion method and detailed in Figure 3. The antibacterial drugs evaluated included azithromycin, clarithromycin, linezolid, mupirocin, doxycycline, minocycline, tetracycline, and tigecycline, all exhibiting varying extents of inhibition zones.

### 4.3. Anti-Pythium Antimicrobial Activity Determined by Broth Microdilution and Gradient Strip Susceptibility Tests

Due to the absence of a specific protocol for *P. insidiosum* susceptibility assays, most broth microdilution tests for this oomycete are conducted following the most recent guidelines of the Clinical and Laboratory Standards Institute’s (CLSI) M38-A2 protocol [62,63], which was initially designed for filamentous fungi. In addition to these standard microdilution methods, susceptibility testing for *P. insidiosum* was conducted using gradient strip tests (Etest^®^ and Liofilchem^®^) (Figure 4). A key distinction, however, is the use of zoospore inocula (Figure 1A), generated in vitro through zoosporogenesis techniques [64], in these tests.

The effectiveness of various antibacterial drugs against *P. insidiosum* has been the subject of several recent research studies revealing varying levels of susceptibility and efficacy. Notably, macrolides (such as azithromycin and clarithromycin) and tetracyclines (including doxycycline, minocycline, and tigecycline) have consistently shown promising in vitro antimicrobial activity against this pathogen, as demonstrated in the studies of Loreto et al. [65], Mahl et al. [66], Worasilchai et al. [9], and Torvorapanit et al. [67], particularly highlighting their low minimum inhibitory concentrations (MICs) in comparison to other evaluated antibacterial drugs, as comprehensively detailed in Table 2. Furthermore, in vitro synergy between tetracyclines and macrolides has also been described, suggesting an enhanced antimicrobial effect when these two classes of antibacterials are combined against *P. insidiosum* [9,67].

Linezolid exhibited similar or slightly higher MICs than macrolides and tetracyclines and was also highlighted as an effective drug inhibiting the in vitro growth of *P. insidiosum* [7,8]. Additionally, Loreto et al. [68] expanded the research to include other oxazolidinones like sutezolid and tedizolid, which demonstrated varying levels of effectiveness. In contrast, aminoglycosides, as studied by Mahl et al. [66], showed less effectiveness due to their higher MICs, a finding further supported by research from Loreto et al. [7], Loreto et al. [6], Worasilchai et al. [9], and Torvorapanit et al. [67].

The research on *P. insidiosum* inhibition by mupirocin and drugs from the pleuromutilin class has revealed some interesting findings. Mupirocin, primarily used as a topical drug, has significantly inhibited *Pythium* growth in vitro [7,8]. Moreover, all evaluated pleuromutilins, common drugs used in veterinary medicine, showed inhibitory activity against this pathogen [68].

Amphenicols, fusidic acid, lincosamides, and streptogramins exhibited higher MICs, indicating a reduced efficacy in inhibiting the in vitro growth of *P. insidiosum*. This variation ranged from moderately elevated MICs to a complete lack of inhibition in some cases [6,7,8].

## 5. Evaluating Protein Synthesis-Inhibiting Antibacterials in Experimental Models of Pythiosis

In the realm of antibacterial treatments, Jesus et al. [72] delved into the in vivo efficacy of azithromycin, clarithromycin, minocycline, and tigecycline against *P. insidiosum*, particularly in the context of subcutaneous pythiosis in a rabbit model. This investigation highlighted that azithromycin, when administered at a dosage of 20 mg/kg/day on a bi-daily schedule, either as a standalone treatment or in conjunction with minocycline at 10 mg/kg/day, led to a significant diminution in microbial load. This reduction was statistically significant and manifested in clinical cures of some animals.

Furthering this line of inquiry, Loreto et al. [73] scrutinized the efficacy of azithromycin in an experimental model involving vascular/disseminated pythiosis in immunocompromised mice. This study specifically assessed the impact of azithromycin administered at 50 mg/kg bi-daily, uncovering a notable decrease in mortality rates. This finding underscores the potential clinical utility of azithromycin in managing this severe variant of pythiosis, with the treatment notably enhancing survival rates to 80% and extending mean survival to 32.4 days.

In 2020, Zimmermann et al. [74] conducted a study to evaluate the effectiveness of minocycline, Pitium-Vac^®^ immunotherapy, and both in treating subcutaneous pythiosis in rabbits. The study found that the combined therapy was significantly more effective in reducing lesion size than using only one or no treatment. Interestingly, one rabbit in the combined treatment group showed complete lesion resolution, which highlights the potential of this approach.

A subsequent study in 2021 by Ahirwar et al. [75] involved the testing of linezolid (0.2%), azithromycin (1%), and tigecycline (1%) in the treatment of induced keratitis in rabbits. The findings of this study revealed that linezolid emerged as the most effective treatment, achieving a 50% success rate and a significant reduction in clinical scores. In contrast, azithromycin and tigecycline demonstrated lower efficacy, with 16.7% and 25% success rates, respectively. Moreover, the study noted adverse reactions in some animals within the azithromycin and tigecycline groups, whereas linezolid was devoid of such adverse effects.

## 6. Exploring the Use of Protein Synthesis-Inhibiting Antibacterials in the Clinical Treatment of Pythiosis

The treatment and management of *Pythium* infections, particularly keratitis, have evolved significantly (Table 3). This evolution is evidenced by a shift from the traditional use of antifungal agents to incorporating antibacterial regimens, especially linezolid and azithromycin [76]. A study by Ramappa et al. [77] exemplifies this shift, where a significant improvement was observed by the fourth day using a combination of topical linezolid, azithromycin, and atropine sulfate, along with oral azithromycin.

The time-related aspects of these treatments are pivotal. Initial antifungal treatments often delayed resolution, necessitating more aggressive interventions such as therapeutic penetrating keratoplasty (TPK). In contrast, Bagga et al. [8] reported a favorable response within 5 to 6 days with the new antibacterial regimen, although a complete cure could take up to 45 days.

Surgical interventions have shown their efficacy in managing severe cases of *Pythium* keratitis and vascular pythiosis [15,18,97]. Studies such as those by Agarwal et al. [81] and Acharya et al. [90] have highlighted that combining TPK with antibacterial therapy and, in some instances, adjunctive procedures like cryotherapy resulted in better outcomes, including a lower recurrence rate and higher rate of globe salvage.

A multidisciplinary approach was more effective in systemic pythiosis, especially in patients with conditions like thalassemia. For instance, Manothummetha et al. [95] reported improved survival rates in such cases with a combination of surgical interventions and a cocktail of antimicrobials.

Collectively, these studies provide insights into the treatment of *Pythium* infections. The timing of treatment initiation and the choice of therapeutic agents are crucial for patient outcomes. However, the data underscore the need for continued research to refine treatment protocols, particularly in understanding the efficacy of antibacterial agents against *Pythium* infections and tailoring treatment plans for different patient demographics.

## 7. Antibacterial Drugs and Pythiosis: Challenges from In Vitro and Experimental Susceptibility to Clinical Insights

Given the structural similarity between *Pythium* species and fungi, initial attempts to standardize susceptibility testing for *P. insidiosum* were based on methods already standardized for fungi, such as broth microdilution assays [62,63] and disk diffusion [98] assays. However, while *Pythium* spp. mycelia can grow on conventional media like Sabouraud dextrose agar, RPMI, and Muller–Hinton broth and agar, zoospores are not produced in these media. To generate zoospores for in vitro susceptibility tests (inoculum), species-specific methodologies that mimic a microrganism’s natural aquatic environment involving water, salts, and plant substrates are required [68]. Additionally, after repeated subculturing in the laboratory, isolates may lose their ability to produce zoospores, which can compromise the reproducibility of susceptibility tests.

In the same way, the first fundamental challenge in the experimental reproduction of pythiosis lies in the necessity to induce the formation of zoospores as the infectious stage of *Pythium* spp. Secondly, the pathogenesis mechanisms of pythiosis, mainly why animals like rabbits—which are not natural hosts—are susceptible to experimental pythiosis while there is no reported success in inducing experimental pythiosis in natural hosts like horses, remain unclear. The disease’s development is presumably tied to an immunological response within the host [26]. Intriguingly, in natural environments where multiple horses are exposed to the same risk factors, only a subset may develop the disease, suggesting individual variations in susceptibility or immune response [99]. Furthermore, reinfection in clinically cured animals, when returned to their original natural environment [100], adds another layer of complexity, indicating a possible lack of lasting immunity against the pathogen. Thirdly, clinical forms of the disease are host-dependent; subcutaneous and ocular forms of pythiosis have only been described in rabbits, with vascular forms only in immunosuppressed mice.

Since the 2010s, many studies have focused on evaluating the susceptibility of *Pythium* species to various classes of antibacterial agents using in vitro susceptibility tests (Table 2) through microdilution, disk diffusion, and gradient strip methods. *Pythium* species have varied in susceptibility to different antibacterial classes inhibiting protein synthesis. Among these antibacterial agents, azithromycin, clarithromycin, linezolid, minocycline, and tigecycline have been evaluated in experimental models of pythiosis [72,74,75].

In experimental models of subcutaneous pythiosis using rabbits, the disease presents as a chronic condition, with subcutaneous lesions expanding over months without evolving into lethal forms. Research by Jesus et al. [72] and Zimmermann et al. [74] demonstrated that tigecycline, minocycline, and azithromycin inhibited lesion progression and achieved clinical cures in some animals, showcasing their potential as effective treatments against *P. insidiosum*. These studies further emphasized that combining or using these drugs with the immunotherapeutic agent PitiumVac^®^ could improve treatment outcomes. In humans, there has been only one reported case of subcutaneous pythiosis, characterized by a right deltoid mass developing over six months, where a regimen incorporating antibacterials was used. In this case, administering itraconazole, azithromycin, and terbinafine led to a gradual lesion regression after three months of follow-up [96].

Ocular pythiosis is clinically distinguished by its rapid onset, often manifesting within ten days or less after exposure to risk factors, and progresses more swiftly than subcutaneous pythiosis. Symptoms resembling a corneal ulcer, including pain, redness, watering, discharge, photophobia, and blurred vision, necessitate immediate diagnosis and intervention to avert severe complications and vision loss [85,97]. The use of antibacterials in treating ocular patients, as outlined in Table 3, underscores their relevance, despite the complexity of discerning their standalone efficacy due to concurrent treatments like therapeutic penetrating keratoplasty (TPK) and other pharmacological drugs that are necessary to manage this clinical condition. However, an experimental study on ocular pythiosis in rabbits assessed the efficacy and safety profiles of azithromycin, linezolid, and tigecycline, suggesting their consideration for trials in human disease [75]. Additionally, review studies propose therapeutic protocols for ocular pythiosis, which now include antibacterial drugs [76,97]. Further research is necessary to evaluate the impact of diagnostic timing and the efficacy of these therapeutic protocols in treating the disease.

Vascular pythiosis is a rare but severe infection that can be life-threatening and often leads to limb loss. Due to its uncommon nature, diagnosis is often delayed, making treatment even more challenging. Amputation is currently the primary course of action, though antimicrobial medications and immunotherapy may be used alongside it [1]. This severity and aggressiveness was observed in an experimental model using immunosuppressed mice, in which the subcutaneous inoculation of zoospores resulted in extremely devastating vascular and systemic impairment, leading to unilateral or bilateral paralysis of the hind limbs and even death of the animals as quickly as 24 to 48 h after disease induction. In this study, azithromycin treatment, initiated 3 h after zoospore inoculation, demonstrated a markedly reduced mortality rate [73].

Azithromycin, doxycycline, and clarithromycin are used as part of the treatment for vascular pythiosis in combination with above-knee amputation, immunotherapy, antifungal drugs, and iron chelator treatments [67,93,94,95] (Table 3). Although radical surgery is considered the primary method of managing this condition, recent multicenter studies by Torvorapanit et al. [67] and Manothummetha et al. [95] described that the addition of azithromycin and doxycycline improved the survival rates in patients with vascular pythiosis who have residual disease, when used in conjunction with surgical and antifungal interventions.

The progression of pythiosis disease is heavily impacted by immunomodulation, which occurs when the pathogen manipulates the immune response of the host to its advantage [26]. This manipulation can often make it difficult to clear the infection, which is why targeted interventions are needed. One potential strategy is the selection of antibacterial drugs that target *P. insidiosum* cells and have a beneficial immunomodulatory effect on the host [10]. Several studies have described the impact of antibacterial drugs on the immune system, including their effects on cellular accumulation, chemotaxis, the microbicidal activity of phagocytic cells, nitric oxide production, cytokine profiles, superoxide anion scavenging, and inflammatory profiles [101,102]. In this context, future research should aim to elucidate the precise mechanisms of action of antibacterial drugs against *P. insidiosum* and consider their potential immunomodulatory role in enhancing strategies to treat pythiosis.

## 8. Conclusions

In conclusion, exploring protein synthesis-inhibiting antibacterials in treating pythiosis offers a nuanced advancement in managing this complex infection. The evidence from in vitro studies, experimental models, and clinical observations suggests a potential benefit of using antibacterials like macrolides, oxazolidinones, and tetracyclines against *P. insidiosum* infections. These findings underscore the necessity of understanding the unique biological characteristics of this pathogen, particularly its distinct susceptibility profile, which sets it apart from fungi.

The clinical application of these findings, however, requires careful consideration. The variability in response to different antibacterials and the potential impact of host-derived sterols on drug efficacy suggests that treatment strategies must be tailored to individual cases. This approach is particularly relevant in severe pythiosis cases, where timely and effective intervention is critical. The promising results from combining surgical interventions with antibacterial therapy in cases like *Pythium* keratitis also point towards a more integrated treatment approach.

## 9. Future Directions

Future research needs to establish standardized protocols for susceptibility testing and further explore the synergistic effects of different antibacterial classes. Additionally, conducting more extensive clinical trials is crucial, especially considering the evolving understanding of pythiosis. For many years, *P. insidiosum* was considered the sole species affecting mammals. However, recent cases have identified other species, such as *P. aphanidermatum* [94,103,104] and *Pythium flevoense* [105], as well as other genera of oomycetes [106,107], as causative agents in mammalian infections. This emerging diversity necessitates a broader scope in the exploration of effective treatments. Such efforts are essential to develop more effective, targeted therapies for pythiosis across its various causative species, ultimately improving patient outcomes and advancing our understanding of this complex and often devastating disease.

## Figures and Tables

**Figure 1 jof-10-00234-f001:**
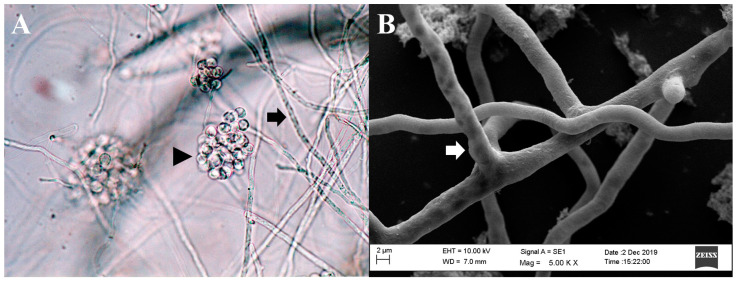
Microscopic morphology of *Pythium insidiosum*. (**A**). Hyphae of *P. insidiosum* (black arrow) and a cluster of encysted zoospores (black arrowhead) (light microscopy, 400× magnification). (**B**). Image from a scanning electron microscope depicting the three-dimensional structure of *P. insidiosum* mycelium (white arrow).

**Figure 2 jof-10-00234-f002:**
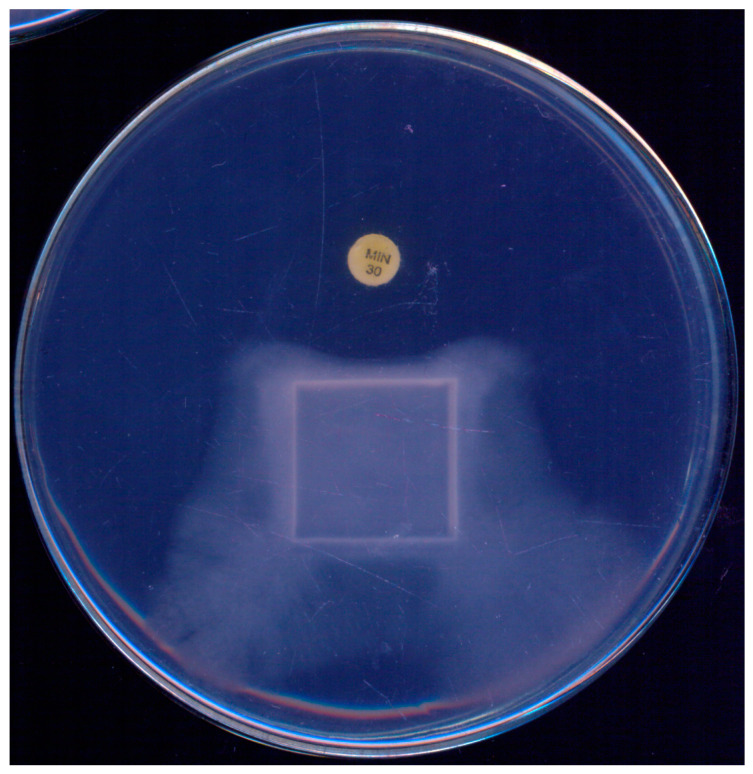
*Pythium insidiosum* growth observed on a single plate after 48 h incubation at 35 °C on Muller–Hinton agar, demonstrating the effect of a minocycline (30 µg) disk. A marked growth inhibition is noticeable in the area surrounding the minocycline disk, illustrating its antibacterial activity against *P. insidiosum*.

**Figure 3 jof-10-00234-f003:**
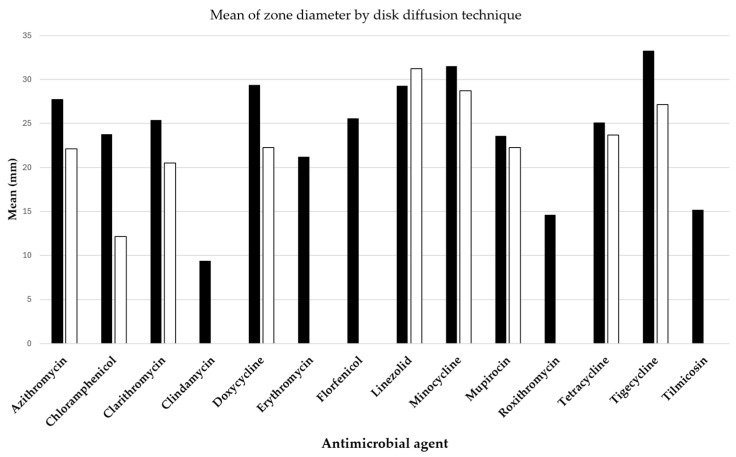
Mean diameters of *Pythium insidiosum* growth inhibition zones around disks containing antibacterial drugs, with black bars representing results from Loreto et al. [7] and white bars indicating findings from Bagga et al. [8].

**Figure 4 jof-10-00234-f004:**
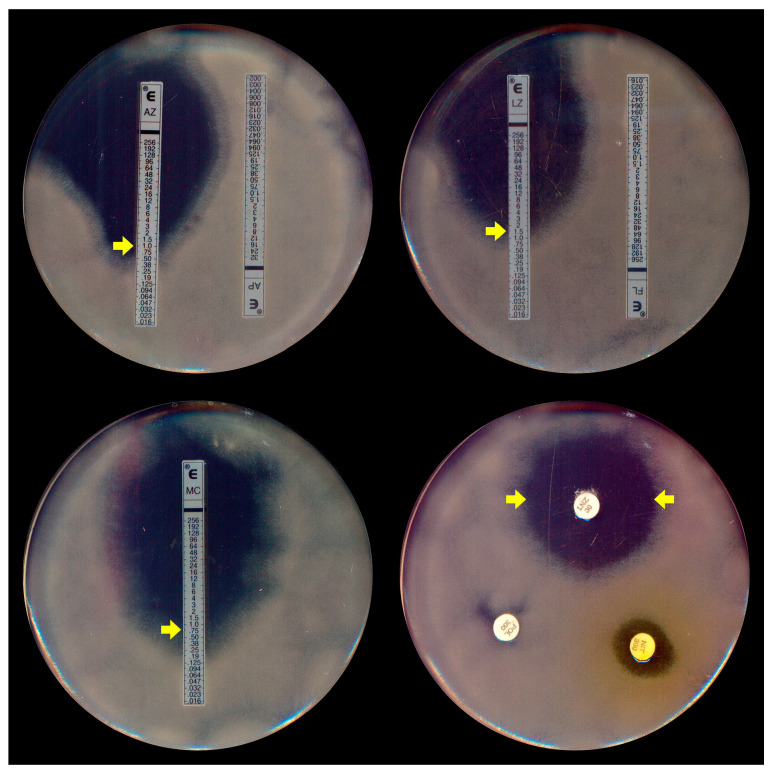
Etest assay (**top left**, **right**, and **bottom left**) demonstrating the elliptical inhibition zones (MIC, indicated by yellow arrows) of *Pythium insidiosum* induced by azithromycin (AZ), linezolid (LZ), and minocycline (MC), respectively. Disk diffusion (**bottom right**) exhibits the halo of linezolid (LNZ) (yellow arrows). Note the absence of inhibition with the antifungal agents amphotericin B (AP) and fluconazole (FL) in the top left and the right plates.

**Table 1 jof-10-00234-t001:** Critical aspects in pythiosis treatment.

Treatment	Key Points	Reference
Surgical Removal	Primary treatment method: Pythiosis is typically treated by surgically removing all affected tissues. Effectiveness It is most effective for superficial and small lesions. It can be challenging to achieve a safe margin to prevent recurrence. In such cases, surgical success rates can drop below 50%. Applicability Surgical removal is often the last therapeutic option. Particularly relevant for humans with vascular and ocular involvement.	[11,13,14,15,16,17,18]
Antifungal drugs	Historical classification of *P. insidiosum*: Initially believed to be a fungus since the 1960s. Use of antifungal drugs: Antifungal drugs were used for treatment based on the initial classification. Limited therapeutic success: These drugs have shown limited success in treating pythiosis. Reclassification as oomycete: Reclassified as an oomycete in the 1970s. This reclassification explained the poor response to antifungals. Reason for poor response to antifungals: Attributed to the absence of ergosterol in *P. insidiosum*’s membrane. Varied treatment outcomes: Despite limited overall success, some cases of canine, equine, and human pythiosis have been cured with antifungal drugs.	[11,19,20,21,22,23,24]
Immunotherapy	Immunotherapy preparation Extracellular or intracellular proteins are obtained by rupturing *P. insidiosum* hyphae using a cell homogenizer, sonicator, vortex shaker, or a combination of these techniques. Antigens for pythiosis immunotherapy preparation can be either cell mass or concentrated soluble antigens. Clinical application and outcome Immunotherapy was used as an adjunctive treatment with antimicrobial drugs or surgery for human pythiosis, achieving cure results of over 80%. In animals, particularly horses, immunotherapy as a primary treatment has achieved cure rates above 70%. Effectiveness was related to the lesion duration, antigen preparation methods, and immunological aspects of host response to the immunotherapeutic antigens. Mechanism of immunotherapy Shifts the cellular response mechanism: (a) Pythiosis: immune response involving eosinophil inflammation and Type 2 T helper cell (Th2) expression; leads to interleukin release, eosinophil, and mast cell mobilization. (b) Immunotherapy: immunotherapeutic antigens promote Type 1 T helper cell (Th1) expression, IL-2, and interferon-γ production; mobilizes T cells and macrophages to destroy *P. insidiosum* cells. Limitations of immunotherapy Does not provide long-term protection against reinfections. Treated hosts remain susceptible to subsequent infections, especially in environments similar to the initial infection.	[12,25,26]
Iodides	Iodides have been suggested as a possible treatment for pythiosis since the disease was first identified. Effectiveness The effectiveness of iodides has been inconsistent. Side effects They can cause serious side effects in some treated animals. Usage in treatment and outcomes Iodides have been used alone or in combination with other therapies. The outcomes of using iodides have varied.	[27,28,29,30,31,32]
Other adjuvant treatments	Several options were described or suggested to complement primary therapies: Neodymium/yttrium–aluminum–garnet (Nd:YAG) laser, photodynamic therapy; iron chelation therapy; plant-derived compounds, like essential oils and tannins, photo-ozone therapy. Clinical application These therapies and compounds have been evaluated through in vitro studies, experimental models, or in animal cases of pythiosis. However, the evidence is based on single studies for most of these suggested treatments. Further clinical evidence is necessary to ascertain these compounds and therapies’ true potential and practical applicability in a clinical setting.	[33,34,35,36,37,38,39,40,41]

**Table 2 jof-10-00234-t002:** Minimum inhibitory concentrations (MICs) of antibacterials inhibiting protein synthesis against *Pythium* spp.

Antibiotic Class	Antimicrobial Agent	MIC Range (Geometric Mean)	Technique	Strain Source (n)	Reference
Aminoglycosides	Amikacin	>32 ^a, 48 h^ >32 ^a, 48 h^ >32 ^a, 48 h^ >32	BMD BMD BMD BMD	Human (17) Environmental (4) Animal (9) Human (8)	[9] [9] [9] [67]
	Gentamicin	32–64 (55.3) ^a, 24 h^ >32 ^a, 48 h^ >32 ^a, 48 h^ 16–>32 (26.91) 16–>32 ^a, 48 h^	BMD BMD BMD BMD BMD	Animal (24) Human (17) Environmental (4) Animal (9) Human (8)	[66] [9] [9] [9] [67]
	Neomycin	32–64 (55.3) ^a^ 32–>32 (32) ^a, 48 h^ >32 ^a, 48 h^ 32–>32 (32) ^a, 48 h^ 32–>32 ^a, 48 h^	BMD BMD BMD BMD BMD	Animal (24) Human (17) Environmental (4) Animal (9) Human (8)	[66] [9] [9] [9] [67]
	Paromomycin	32–64 (49.3) ^a, 24 h^	BMD	Animal (25)	[66]
	Streptomycin	32–64 (50.7) ^a, 24 h^ 16–>32 (22.63) ^a, 48 h^ >32 ^a, 48 h^ 16–>32 (26.91) ^a, 48 h^ 16–>32 ^a, 48 h^	BMD BMD BMD BMD BMD	Animal (24) Human (17) Environmental (4) Animal (9) Human (8)	[66] [9] [9] [9] [67]
	Tobramycin	>1024 ^a, 24 h^ >32 ^a, 48 h^ >32 ^a, 48 h^ >32 ^a, 48 h^ >32	BMD BMD BMD BMD BMD	Animal (28) Human (17) Environmental (4) Animal (9) Human (8)	[7] [9] [9] [9] [67]
Amphenicols	Chloramphenicol	4–>256 (23.1) ^a, 24 h^ and 4–>256 (52.5) ^a, 48 h^ 2–>256 (25.6) ^a, 24 h^ and 8–>256 (53.8) ^a, 48 h^ 16.00–256 ^48 h^	BMD Etest Etest	Animal (28) Animal (28) Human (38)	[7] [7] [8]
	Florfenicol	8–>256 (25.1) ^a, 24 h^ and 16–>256 (50.2) ^a, 48 h^	BMD	Animal (28)	[7]
Fusidanes	Fusidic acid	>256 ^a, 24 h^	BMD and Etest	Animal (28)	[7]
Lincosamides	Clindamycin	4–>256 (16) ^a, 24 h^ and 4–>256 (26.9) ^a, 48 h^ 2–256 (7.6) ^a, 24 h^ and 2–>256 (14.5) ^a, 48 h^	BMD Etest	Animal (28) Animal (28)	[7] [7]
	Lincomycin	>256 ^a, 24 h^	BMD	Animal (28)	[7]
Macrolides and ketolides	Azithromycin	2–32 (4.57) ^a, 24 h^ and 0.5–2 (1.11) ^b, 24 h^ 1–8 (2.9) ^a, 24 h^ and 1–16 (3.9) ^a, 48 h^ 0.03–4 (0.7) ^a, 24 h^ and 0.03–16 (1.0) ^a, 48 h^ 0.02–32 ^48 h^ 1–32 (6.96) ^a, 48 h^ and 0.5–8 (1.78) ^b, 48 h^ 2–16 (4.68) ^a,48 h^ 1–4 (3.13) ^a,48 h^ 2–16 (4.76) ^a, 48 h^ 2–8 (2.72) ^a, 48 h^ 8–64 (18.38) ^a, 24 h^ and 1–8 (2.30) ^b, 24 h^ 2–4 ^a, 48 h^	BMD BMD Etest Etest BMD BMD BMD BMD BMD BMD BMD	Animal (26) Animal (28) Animal (28) Human (38) Animal (30) Animal (21) Human (17) Environmental (4) Animal (9) Animal (20) Human (8)	[65] [7] [7] [8] [68] [69] [9] [9] [9] [70] [67]
		2–32 (7.46) ^a, 48 h^	BMD	Animal (20)	[71]
	Clarithromycin	0.5–8 (1.53) ^a, 24 h^ and 0.125–1 (0.49) ^b, 24 h^ 0.25–8 (1.8) ^a, 24 h^ and 0.25–8 (3.1) ^a, 48 h^ 0.5–16 (2.4) ^a, 24 h^ and 0.5–32 (3.9) ^a, 48 h^ 0.05–4 ^48 h^ 0.5–64 (4.49) ^a, 48 h^ and 0.5–8 (1.19) ^b, 48 h^ 0.125–8 (1.33) ^a, 48 h^ 2 (2) ^a, 48 h^ 0.125–2 (1.0) ^a, 48 h^ 0.125–2 ^a, 48 h^	BMD BMD Etest Etest BMD BMD BMD BMD BMD	Animal (26) Animal (28) Animal (28) Human (38) Animal (30) Human (17) Environmental (4) Animal (9) Human (8)	[65] [7] [7] [8] [68] [9] [9] [9] [67]
	Erythromycin	2–32 (7.58) ^a, 24 h^ and 0.5–4 (1.61) ^b, 24 h^ 1–32 (7.7) ^a, 24 h^ and 2–64 (15.5) ^a, 48 h^	BMD BMD	Animal (26) Animal (28)	[65] [7]
	Josamycin	2–64 (16) ^a, 48 h^ and 0.5–16 (2.33) ^b, 48 h^	BMD	Animal (30)	[68]
	Roxithromycin	2–128 (9.7) ^a, 24 h^ and 4–128 (20.6) ^a, 48 h^	BMD	Animal (28)	[7]
	Telithromycin	0.5–4 (1.15) ^a, 48 h^	BMD	Animal (20)	[71]
	Tilmicosin	4–128 (27.6) ^a, 24 h^ and 8–128 (42.8) ^a, 48 h^	BMD	Animal (28)	[7]
Oxazolidinones	Linezolid	1–32 (5.6) ^a, 24 h^ and 4–32 (8.8) ^a, 48 h^ 0.5–8 (1.7) ^a, 24 h^ and 0.5–8 (2.0) ^a, 48 h^ 0.75–32 ^48 h^ 1–64 (13.30) ^a, 48 h^ and 1–32 (4.11) ^b, 48 h^ 4–32 (8.33) ^a, 48 h^ 4–16 (9.51) ^a, 48 h^ 4–8 (5.44) ^a, 48 h^ 4–8 ^a, 48 h^	BMD Etest Etest BMD BMD BMD BMD BMD	Animal (28) Animal (28) Human (38) Animal (30) Human (17) Environmental (4) Animal (9) Human (8)	[7] [7] [8] [68] [9] [9] [9] [67]
	Sutezolid	4–64 (7.46) ^a, 48 h^ and 1–4 (2.24) ^b, 48 h^	BMD	Animal (30)	[68]
	Tedizolid	>32 ^48 h^	MIC Test Strip	Animal (30)	[68]
Pleuromutilins	Retapamulin	0.25–32 (1.45) ^a, 48 h^ and <0.125–32 (0.15) ^b, 48 h^	BMD	Animal (30)	[68]
	Tiamulin	2–64 (16.37) ^a, 48 h^ and 1–8 (3.21) ^b, 48 h^	BMD	Animal (30)	[68]
	Valnemulin	0.25–16 (2.09) ^a, 48 h^ and <0.125–4 (0.22) ^b, 48 h^	BMD	Animal (30)	[68]
Pseudomonic Acids	Mupirocin	2–32 (3.2) ^a, 24 h^ and 2–32 (6.9) ^a, 48 h^ 0.125–2 (0.6) ^a, 24 h^ and 0.125–4 (1.0) ^a, 48 h^ 1–8 (2.49) ^a, 48 h^ 0.06–1.50 ^48 h^	BMD Etest BMD Etest	Animal (28) Animal (28) Animal (21) Human (38)	[7] [7] [69] [8]
Streptogramins	Synercid	0.5–>32 (5.8) ^a, 24 h^ and 0.5–>32 (6.9) ^a, 48 h^	Etest	Animal (28)	[7]
Tetracyclines and Glycylcyclines	Doxycycline	0.5–8 (1.75) ^a, 24 h^ and 0.125–1 (0.35) ^b, 24 h^ 1–8 (3.3) ^a, 24 h^ and 2–16 (6.4) ^a, 48 h^ 1–8 (2.3) ^a, 24 h^ and 2–16 (5.8) ^a, 48 h^ 0.13–12 ^48 h^ 1–16 (3.69) ^a, 48 h^ 4–8 (4.76) ^a, 48 h^ 1–16 (3.43) ^a, 48 h^ 1–4 ^a, 48 h^	BMD BMD Etest Etest BMD BMD BMD BMD	Animal (26) Animal (28) Animal (28) Human (38) Human (17) Environmental (4) Animal (9) Human (8)	[65] [7] [7] [8] [9] [9] [9] [67]
	Minocycline	0.125–2 (0.39) ^a, 24 h^ and 0.06–0.5 (0.08) ^b, 24 h^ 0.125–4 (0.9) ^a, 24 h^ and 0.25–4 (1.6) ^a, 48 h^ 0.06–4 (0.2) ^a, 24 h^ and 0.06–4 (0.4) ^a, 48 h^ 0.02–4 ^48 h^ 1–4 (1.63) ^a, 48 h^ 2 (2) ^a, 48 h^ 0.25–4 (1.08) ^a, 48 h^ 0.25–2 ^a, 48 h^	BMD BMD Etest Etest BMD BMD BMD BMD	Animal (26) Animal (28) Animal (28) Human (38) Human (17) Environmental (4) Animal (9) Human (8)	[65] [7] [7] [8] [9] [9] [9] [67]
	Oxytetracycline	2–32 (7.38) ^a, 24 h^ and 1–2 (1.57) ^b, 24 h^	BMD	Animal (26)	[65]
	Tetracycline	2–32 (5.96) ^a, 24 h^ and 0.5–2 (1.2) ^b, 24 h^ 1–32 (7.4) ^a, 24 h^ and 4–32 (16) ^a, 48 h^ 0.19–24 ^48 h^	BMD BMD Etest	Animal (26) Animal (28) Human (38)	[65] [7] [8]
	Tigecycline	0.25–2 (0.9) ^a, 24 h^ 0.25–4 (1.3) ^a, 24 h^ and 0.5–4 (2) ^a, 48 h^ 0.03–4 (0.2) ^a, 24 h^ and 0.03–4 (0.3) ^a, 48 h^ 0.02–1.50 ^48 h^ 1–4 (1.57) ^a, 48 h^ 2 (2) ^a, 48 h^ 0.5–2 (1.08) ^a, 48 h^ 0.5–2 ^a, 48 h^	BMD BMD Etest Etest BMD BMD BMD BMD	Animal (24) Animal (28) Animal (28) Human (38) Human (17) Environmental (4) Animal (9) Human (8)	[66] [7] [7] [8] [9] [9] [9] [67]

^a^, 100% growth inhibition; ^b^, 50% growth inhibition; ^24 or 48 h^, time of MICs determination; BMD, Broth microdilution; n, number of strains evaluated.

**Table 3 jof-10-00234-t003:** Summary of clinical cases detailing the efficacy of antibacterial drugs in treating *Pythium* infections.

Case Details	Treatment Regimen	Outcome and Notable Points	Reference
Ocular Pythiosis			
A 42-year-old woman with presumptive *Pythium* keratitis.	Topical linezolid, azithromycin, and atropine sulfate; oral azithromycin.	Significant improvement by the fourth day. The infection completely resolved within three weeks.	[77]
A 30-year-old man with ocular pythiosis.	Before surgery: ophthalmic moxifloxacin, amikacin, vancomycin, and penetrating keratoplasty. After surgery: topical voriconazole, natamycin, voriconazole, liposomal amphotericin B, chlorhexidine, caspofungin, and cyclosporine. After *P. insidiosum* diagnosis: oral minocycline and terbinafine. Intracameral minocycline (during the third keratoplasty)	Initially, the condition worsened and required multiple surgeries. After a 45-day hospital stay, the patient was discharged and prescribed oral minocycline, cyclosporine, and ofloxacin eye drops. Two months later, the patient was infection free following the oral minocycline treatment.	[78]
Study of *P. insidiosum* keratitis in 114 LV Prasad Eye Institute patients.	Topical natamycin, voriconazole, and oral ketoconazole or itraconazole, used until 2016, were replaced with topical linezolid, azithromycin, and oral azithromycin.	The initial standard treatment has shown varied responses. A new regimen with antibacterial drugs has resulted in a lower rate of TPK and a higher proportion of healed ulcers. The response has been favorable within 5 to 6 days, although a complete cure may take 30 to 45 days.	[8]
A 7-year-old boy with *P. insidiosum* keratitis.	Initial treatment: topical natamycin, atropine, voriconazole, and oral analgesics. Second treatment: topical azithromycin, voriconazole, oral azithromycin, cyanoacrylate adhesive, and a bandage contact lens.	There was no improvement with the initial treatment. Improvement was noted after changing the treatment to azithromycin and voriconazole. Oral azithromycin was discontinued after four weeks, and vision improved to perception of hand movement after 13 weeks.	[79]
A case of keratitis, coinfected with *P. insidiosum* and *Acanthamoeba*.	Initial treatment: natamycin and moxifloxacin. After diagnosis: topical therapy supplemented with polyhexamethylene biguanide (PHMB), topical and oral linezolid, intrastromal voriconazole, topical and oral azithromycin, TPK, topical and oral prednisolone, topical fluorometholone.	The treatment was deemed successful four months postoperatively, evidenced by the clear graft.	[80]
Study on 46 patients with *Pythium* keratitis to compare various treatments.	(a) Medical management (MM) treatment, consisting of topical azithromycin and linezolid along with oral azithromycin, was applied to 1 eye upon initial presentation. (b) TPK was the primary surgical intervention for 42 eyes. (c) A surgical adjunct approach (SA), integrating TPK with cryotherapy and/or alcohol, was employed for 3 eyes. (d) Following TPK, 8 eyes received adjunct medical management (MA) with antibacterial drugs to prevent recurrence.	The most effective treatment for *Pythium* keratitis was TPK and, in severe cases, evisceration. Adjunctive procedures during TPK showed benefits with a lower risk of recurrence and could be considered routine care. Despite a high recurrence rate, 39 out of 46 eyes were anatomically salvaged.	[81]
*P. insidiosum* keratitis in a 20-year-old Japanese man.	Initial treatment: topical pimaricin, voriconazole, and intravenous liposomal amphotericin B. Treatment shifted to topical minocycline, chloramphenicol, and oral linezolid.	The condition worsened despite initial treatment. Significant improvement in keratitis with the new regimen. Developed corneal perforation, necessitating therapeutic penetrating keratoplasty. No recurrence of infection in the 11 months following surgery.	[82]
Study on two patients with *P. insidiosum* keratitis.	Case 1: 45-year-old male. Initial treatment: topical linezolid and azithromycin. Progression of infection led to TPK with cryotherapy and alcohol swabbing. Post-surgery: continued topical treatments and oral azithromycin. Case 2: 62-year-old male. Initial treatment: topical linezolid, azithromycin, oral azithromycin, and TPK with alcohol application and cryotherapy. Postoperative management: reduced topical antibiotics, removal of loose sutures, introduction of loteprednol and carboxymethylcellulose eye drops.	Both patients were effectively cured *of P. insidiosum* keratitis. Careful postoperative management was crucial for successful outcomes.	[83]
Retrospective analysis of 112 patients with *P. insidiosum* keratitis.	Sixty-nine patients were treated with topical linezolid, azithromycin, and oral azithromycin. Excluded patients with severe corneal thinning, perforations, limbal/scleral involvement, or endophthalmitis who underwent early TPK.	In total, 55.1% (38 of 69 eyes) responded to medical therapy; 34.3% required cyanoacrylate glue for tectonic support; and 44.9% (31 of 69 eyes) underwent TPK. Post-TPK: 29% of grafts remained clear, and 70.9% experienced graft failure. No recurrence of infection was observed.	[84]
A study involving 30 patients with *Pythium* keratitis.	Before culture results: topical natamycin (7 patients), natamycin and voriconazole (15), natamycin and itraconazole (8). After culture results: topical linezolid (11), linezolid and azithromycin (19). TPK in 63.3% of patients. Post-TPK: topical linezolid and azithromycin.	A total of seven patients healed with medical treatment, nineteen healed with TPK, and four were lost to follow-up.	[85]
Three cases of *P. insidiosum* keratitis in adults from China.	Case 1: 45-year-old female. Treatments: topical/systemic fluconazole, levofloxacin, cefminox sodium, intracameral fluconazole injection, lamellar keratoplasty, post-surgery amphotericin B. Case 2: 51-year-old female. Treatments: topical levofloxacin, cefminox sodium, voriconazole, TPK, intracameral fluconazole. Case 3: 55-year-old male. Treatments: topical/systemic antibiotics (ornidazole, tobramycin, vancomycin, natamycin, fluconazole), excision of pterygium, TPK, antiamebic therapy (chlorhexidine), additional voriconazole, intracameral amphotericin B.	Case 1: enucleation of right eye due to ineffective antifungal therapy and increasing infiltrate with hypopyon. Post-enucleation adjusted treatment with linezolid and azithromycin, no recurrence observed. Case 2: enucleation on day 28 post-exposure due to progressive, unresponsive infiltrates. Case 3: enucleation due to infection spreading to adjacent sclera and progressing to endophthalmitis. Despite treatments, all cases resulted in enucleation due to uncontrolled infection.	[86]
Retrospective review of medical records from 2006 to 2019 of patients diagnosed with *Pythium* keratitis.	Most were treated with at least two topical antifungal agents: natamycin and voriconazole (14 eyes, 53.8%), topical natamycin, and amphotericin B (11 eyes, 42.3%). Topical antibiotics: moxifloxacin (15 eyes), azithromycin (5 eyes), linezolid (3 eyes). Oral antifungals: terbinafine and itraconazole. TPK in 21 eyes.	Despite drug treatments, the infection progressed in 24 out of 26 cases (92.3%). After the first TPK, 6 out of 21 eyes (28.6%) showed improvement without additional surgery. However, 15 out of 21 eyes (71.4%) experienced a recurrence. Globe salvage was achieved in 11 eyes (42.3%), while enucleation was necessary in 15 eyes (57.7%).	[87]
A retrospective study on 21 cases of *Pythium* keratitis.	Topical linezolid and azithromycin, and oral linezolid.	Keratitis resolution and corneal scarring in 73.68% of cases (14 out of 19). TPK was performed in four cases due to lack of response, large infiltrates, or worsening conditions. All corneal grafts in these cases failed. Two patients underwent successful optical penetrating keratoplasty and endothelial keratoplasty. One patient with a large infiltrate and extensive intraocular infection underwent evisceration.	[88]
A 9-year-old boy with *P. insidiosum* keratitis.	Initial treatment: topical antifungals (natamycin and itraconazole), cycloplegic homatropine, and oral diclofenac. After deterioration: shift to antibacterial agents (topical linezolid and azithromycin). Intervention with cyanoacrylate glue and a bandage contact lens due to rapid progression of infiltrate and early corneal melting.	Within two weeks, signs of healing were evident; visual acuity improved to 5/60. After one month, best-corrected visual acuity (BCVA) improved to 6/12. Using cyanoacrylate glue, which has antibacterial properties, enhanced the treatment’s efficacy.	[89]
Retrospective study on 16 patients with *P. insidiosum* keratitis.	Before and after TPK: topical linezolid and azithromycin, homatropine, and oral azithromycin. After TPK: prednisolone acetate. Intracameral linezolid was used during surgery.	Nine patients (56.25%) experienced a relapse, which was managed through repeat keratoplasty, cryotherapy, or additional intracameral linezolid. Globe was salvaged in 14 out of 16 patients (87.5%).	[90]
A 44-year-old male patient with HIV and acute retinal necrosis developed *P. insidiosum* keratitis.	Initially, topical natamycin, voriconazole, cycloplegic homatropine TDS, antiglaucoma timolol, oral diclofenac with serratiopeptidase, and pantoprazole. The treatment was then shifted to topical linezolid and azithromycin, with adjuvant drugs continuing.	By the fifth week, there was an improvement, and a complete resolution was achieved after seven weeks. No recurrence was noted during the two-month follow-up.	[91]
Retrospective analysis of TPK in patients with *P. insidiosum* keratitis.	Preoperatively, topical linezolid and azithromycin. Postoperatively, topical linezolid and azithromycin, as well as oral azithromycin, are used.	Out of 238 cases, 50 cases met the inclusion criteria. The study found that patients with *P. insidiosum* keratitis usually require TPK despite being treated with antibacterial drugs. However, these grafts’ anatomical and functional outcomes are significantly better than antifungal regimens. Moreover, lower recurrence rates were observed in cases treated with TPK and antibacterial drugs.	[92]
Vascular Pythiosis			
Study on two cases of intra-abdominal pythiosis treated with surgical interventions and adjunctive antibacterial therapy.	Case 1: 37-year-old male with beta-thalassemia. Initial treatment: above-knee amputation, itraconazole, and PIV immunotherapy (mixture of extracellular and intracellular proteins). Following infection persistence: oral azithromycin and itraconazole. After recurrent abdominal pain and aneurysm progression: switched to oral doxycycline and clarithromycin. Case 2: 48-year-old male with thalassemia major. Initial treatment: above-knee amputation, oral itraconazole, and immunotherapy. Following new aneurysm development and high serum BG levels: supplemented with oral azithromycin. Persistent symptoms: switched to oral voriconazole and doxycycline. Later transitioned back to itraconazole and continued with azithromycin and doxycycline.	Case 1 and 2: remained well at 64 weeks post-diagnosis. Both cases involved combined medical therapies. These cases highlight the complexity and adaptability required in managing severe pythiosis, especially in patients with underlying conditions like thalassemia.	[93]
A 47-year-old Thai woman with beta-thalassemia/hemoglobin E presented with acute arterial insufficiency in both legs associated with *Pythium aphanidermatum* infection.	The patient received treatment with itraconazole, terbinafine, azithromycin, doxycycline, and the iron chelator deferoxamine. Radical surgery was not feasible, and immunotherapy with a vaccine was unavailable.	Regrettably, the patient passed away from uncontrolled sepsis two weeks following treatment with itraconazole, terbinafine, azithromycin, doxycycline, and the iron chelator deferoxamine.	[94]
Multicenter, prospective cohort study on vascular pythiosis patients with underlying thalassemia.	Combination of radical surgery and antimicrobial treatment, including azithromycin, doxycycline, and antifungal agents	Four of the eight patients evaluated had residual disease postoperatively; two were managed with antimicrobials alone. One case required a second surgery; one patient succumbed five months later.	[67]
A clinical trial with 40 patients with vascular pythiosis.	Surgery and a combination of itraconazole, doxycycline, and azithromycin.	At the 6-month follow-up of this study, out of the total participants, 3 patients (7.5%) died due to complications including disseminated pythiosis and infections. Out of the cohort, 26 patients showed no residual disease post-operatively. In contrast, 14 patients (35%) had residual disease; among them, 1 patient (7.1%) died, while 13 patients survived.	[95]
Cutaneous/Subcutaneous Pythiosis			
A 26-year-old pregnant woman with subcutaneous pythiosis.	Painkillers, oral doxycycline, and cloxacillin were replaced with itraconazole, azithromycin, and terbinafine.	Despite initial treatments, the condition persisted. However, after being diagnosed with pythiosis and switching treatments, the lesion regressed gradually and was successfully treated.	[96]

TPK, Therapeutic penetrating keratoplasty.

## Data Availability

Data are contained within the article.

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
