# Peer review of "Treating Pythiosis with Antibacterial Drugs Targeting Protein Synthesis: An Overview"

_jof, 2024, doi:10.3390/jof10040234_

Round 1

Reviewer 1 Report

This is an interesting and well-presented manuscript written by experts in the field of pythiosis antimicrobial therapy, as evidenced by their several publications dating from more than a decade. Nevertheless, the title does not seem to be adequate as it suggest the discovery of a solution, or nearly (“shifting paradigms”), for the treatment of the disease and a comparison with previous paradigms or treatments, while the truth is that no much advances have been made in pythiosis treatment in many years. 

There is a nice recent (2022) review on antimicrobial therapy for human phytiosis (doi: 10.3390/antibiotics11040450.) that is largely based on in vitro studies published by the authors of the present manuscript. The mentioned review gives a concise picture of clinical and pathogenic aspects and then focuses on current and future treatment options. The present MS “aims to explore the activity of protein synthesis-inhibiting antibacterials against P.insidiosum, delving into their efficacy through a comprehensive analysis of in vitro susceptibility data, experimental infection models and clinical case studies”. However, there is a quite extensive description of these issues but, to my point of view, there is not such a comprehensive analysis that could add new insight into the treatment of pythiosis. As with the title, the objective anticipates at least some news that I do not see in the conclusions.  

There is also a more extensive review from 2020 that updates diagnostic and management aspects of pythiosis. Eight articles of the authors of the present MS are cited.

So, rather than attempt to perform a description using the same literature and achieving the same conclusions as other recent review articles, I would suggest the authors to take all the valuable data generated by themselves and make an original contribution to the scientific community. For instance, try to correlate in vitroresults with clinical data (very well detailed in the tables 1 and 2), and focus on the contradictory in vitro data and in vivo outcomes in an attempt to advance in the establishment of susceptibility testing standardized protocols and breakpoints. This approach could be enriched by explaining more in deep the different methodologies for in vitro susceptibility testing enumerated in section 3. On the other hand, while pretending a comprehensive review, the authors do not refer to immunomodulating activity of the antibacterials.  

Another interesting contribution could be made by means of a systematic review or a meta-analysis, although there are probably not enough reported cases and trials to apply these methodologies.

Minor comments:

-       There is a newer CLSI document than the one cited. It is the 3rd edition and dates from 2017. This should be revised and updated.

-       Figure 3 shows in vitro resistance to Amphotericin B. However, successful treatments have been reported (doi: 10.1111/j.1600-0560.1993.tb00654.x).

-       A mention should be made of the differences between animal and clinical cases. For instance, table 1 refers mostly to animal isolates in vitro susceptibility while table 2 refers only to human cases.

-       Table 2. I would suggest to separate the different clinical presentations as they may respond very differently to antimicrobials according to the site of infection and severity. Of note, most cases correspond to keratitis, that would respond to topical treatment. Also, the discussion should acknowledge clinical failure with the promising antimicrobials and cure cases with surgery/antifungal, probably independent of antibacterial treatment. 

-       Again, as with the title and the aims of the review, section 5 refers to “evolving” landscape in pythiosis, and they even say that treatment and management have “evolved significantly”, whereas the reviewed case reports, clinical trials and retrospective studies do not show such a significant change. If table 2 was ordered by year of publication (which is almost done) we can see that from 2017 to 2023 there is no a clear change in management practices. In fact, the first described case is a keratitis case successfully treated with linezolid and azithromycin.

Author Response

Dear Reviewer

We acknowledge the receipt of your review and thank you for the time spent evaluating our manuscript. Your comments are noted and appreciated for the critical insights they offer.

The responses to your observations have been compiled and can be found below. These points have been addressed and marked within the manuscript accordingly. The feedback has been instrumental in guiding the revisions, and we have tried to incorporate the necessary changes.

Once again, thank you for your contribution towards improving our work.

Q1. The title does not seem to be adequate as it suggest the discovery of a solution, or nearly (“shifting paradigms”), for the treatment of the disease and a comparison with previous paradigms or treatments, while the truth is that no much advances have been made in pythiosis treatment in many years.

We understand your concern that the original title might have implied a significant paradigm shift or a breakthrough in the treatment of pythiosis, which does not align with the current state of advancements in this field.

We have revised the title to "Treating pythiosis with antibacterials targeting protein synthesis: An overview" to address this issue. This new title more accurately reflects the content and objective of our manuscript, focusing specifically on the use of antibacterials that inhibit protein synthesis in the treatment of pythiosis without suggesting a major paradigm shift. This alteration aligns better with our research's actual advancements and scope.

Q2. Are the conclusion supported by results? As with the title and the aims of the review, authors refers to “evolving” landscape in pythiosis, and they even say that treatment and management have “evolved significantly”, they mention a "shift of paradigm" whereas the reviewed case reports, clinical trials and retrospective studies do not show such a significant change.

We appreciate the insightful observations regarding the language used in our manuscript, particularly concerning the 'evolving landscape' and 'significant evolution' in treating and managing pythiosis. Upon reflection, we acknowledge that our choice of words may have inadvertently suggested a more dramatic advancement in the field than the current evidence from case reports, clinical trials, and retrospective studies robustly supports. The first paragraph of the conclusion was rewritten to demonstrate these alterations.

 Q3. Does this article provide a relevant contribution to the scientific discussion of this topic?

There are two recent reviews about the same topic and both of them discuss relevant articles published by the authors of the present manuscript, who are, without doubt, experts in the field of antimicrobial treatment of pythiosis. One is a nice review from 2022 on antimicrobial therapy for human phytiosis (doi: 10.3390/antibiotics11040450.) that is largely based on in vitro studies published by the authors of the present manuscript. The mentioned review gives a concise picture of clinical and pathogenic aspects and then focuses on current and future treatment options.

The present MS “aims to explore the activity of protein synthesis-inhibiting antibacterials against P.insidiosum, delving into their efficacy through a comprehensive analysis of in vitro susceptibility data, experimental infection models and clinical case studies”. However, there is a quite extensive description of these issues but, to my point of view, there is not such a comprehensive analysis that could add new insight into the treatment of pythiosis. As with the title, the objective anticipates at least some news that I do not see in the conclusions. There is also a more extensive review from 2020 that updates diagnostic and management aspects of pythiosis. Eight articles of the authors of the present MS are cited.

Thank you for your perspective. When considering the research on pythiosis treatment from a five-year viewpoint, it might appear that there haven't been significant advancements, possibly due to the initiation of the clinical use of antibacterials like linezolid and azithromycin, which have shown very good therapeutic responses in some studies. However, when examining the evolution of pythiosis treatment over the last 20 years, I do recognize that the introduction of antibacterial drugs, following in vitro and experimental testing, into human clinical practice has been a major advancement.

The cited article (doi: 10.3390/antibiotics11040450, A Review: Antimicrobial Therapy for Human Pythiosis), along with other excellent publications from 2020 (doi´s: 10.1016/j.heliyon.2020.e03737; 10.7717/peerj.8555; 10.1016/j.heliyon.2020.e03737), serve as excellent examples of the recognition of these important research efforts that culminated in the clinical use of these antibacterials to treat pythiosis. This is precisely why they are also referenced in our manuscript.

However, it's important to note that the focus of these works differs from that of our manuscript despite some overlap in the information presented. Some studies focus solely on susceptibility methods, others describe the same antimicrobials mentioned in our manuscript but only in the context of human pythiosis, and there are also reviews covering the various treatments used for pythiosis.

Our manuscript aims to provide readers with a comprehensive compilation of data specifically regarding the use of antibacterials. This is why we have structured the text to evolve from in vitro tests to experimental studies and then to clinical applications. However, it's worth mentioning that a thorough conceptual exploration of all these topics could indeed yield several individual manuscripts. We attempted to condense the information to provide a succinct overview of the evolution of introducing antibacterials in the treatment of pythiosis. In this process, understandably, some gaps may emerge, and certain significant details may not receive the emphasis they deserve.

 Q4 - Major comments -

 This is an interesting and well-presented manuscript written by experts in the field of pythiosis antimicrobial therapy, as evidenced by their several publications dating from more than a decade. Nevertheless, the title does not seem to be adequate as it suggest the discovery of a solution, or nearly (“shifting paradigms”), for the treatment of the disease and a comparison with previous paradigms or treatments, while the truth is that no much advances have been made in pythiosis treatment in many years.

Answered in Q1.

 There is a nice recent (2022) review on antimicrobial therapy for human phytiosis (doi: 10.3390/antibiotics11040450.) that is largely based on in vitro studies published by the authors of the present manuscript. The mentioned review gives a concise picture of clinical and pathogenic aspects and then focuses on current and future treatment options. The present MS “aims to explore the activity of protein synthesis-inhibiting antibacterials against P.insidiosum, delving into their efficacy through a comprehensive analysis of in vitro susceptibility data, experimental infection models and clinical case studies”. However, there is a quite extensive description of these issues but, to my point of view, there is not such a comprehensive analysis that could add new insight into the treatment of pythiosis. As with the title, the objective anticipates at least some news that I do not see in the conclusions. 

Answered in Q3.

 There is also a more extensive review from 2020 that updates diagnostic and management aspects of pythiosis. Eight articles of the authors of the present MS are cited.

Answered in Q3. 

So, rather than attempt to perform a description using the same literature and achieving the same conclusions as other recent review articles, I would suggest the authors to take all the valuable data generated by themselves and make an original contribution to the scientific community.

For instance, try to correlate in vitro results with clinical data (very well detailed in the tables 1 and 2), and focus on the contradictory in vitro data and in vivo outcomes in an attempt to advance in the establishment of susceptibility testing standardized protocols and breakpoints. This approach could be enriched by explaining more in deep the different methodologies for in vitro susceptibility testing enumerated in section 3. On the other hand, while pretending a comprehensive review, the authors do not refer to immunomodulating activity of the antibacterials. 

In response to your suggestions, we are introducing a new session titled " Antibacterial drugs and pythiosis: challenges from in vitro and experimentally susceptibility to clinical insights " to address your critical points.

This session will briefly focus on the standardization challenges of in vitro susceptibility tests, their correlation with clinical data, discrepancies between in vitro and in vivo results, and immunomodulating activities of antibacterials, recognizing their significance in interpreting drug efficacy in vivo.

  Another interesting contribution could be made by means of a systematic review or a meta-analysis, although there are probably not enough reported cases and trials to apply these methodologies.

 Q5 - Minor comments:

-       There is a newer CLSI document than the one cited. It is the 3rd edition and dates from 2017. This should be revised and updated.

Thank you for the observation. The text has been updated.

 -       Figure 3 shows in vitro resistance to Amphotericin B. However, successful treatments have been reported (doi: 10.1111/j.1600-0560.1993.tb00654.x).

The new "2. Overview" section states that antifungals have reports of effectiveness in some cases. In the example provided, the isolate was resistant.

 -       A mention should be made of the differences between animal and clinical cases. For instance, table 1 refers mostly to animal isolates in vitro susceptibility while table 2 refers only to human cases.

-       Table 2. I would suggest to separate the different clinical presentations as they may respond very differently to antimicrobials according to the site of infection and severity. Of note, most cases correspond to keratitis, that would respond to topical treatment.

The data in the table were separated by the clinical form of the disease and reorganized by year of publication and alphabetical order, depicting sources (animal, human). 

Also, the discussion should acknowledge clinical failure with the promising antimicrobials and cure cases with surgery/antifungal, probably independent of antibacterial treatment.

New critical section described the challenges.

  -       Again, as with the title and the aims of the review, section 5 refers to “evolving” landscape in pythiosis, and they even say that treatment and management have “evolved significantly”, whereas the reviewed case reports, clinical trials and retrospective studies do not show such a significant change.

The expression “evolving landscape” was replaced.

 If table 2 was ordered by year of publication (which is almost done) we can see that from 2017 to 2023 there is no a clear change in management practices. In fact, the first described case is a keratitis case successfully treated with linezolid and azithromycin.

The year of publication was updated.

Reviewer 2 Report

This is a nice review. However, it needs some refinement to improve the paper.

The immunotherapies (lines 42-44) need to be expandded. Details of what they are and their effect need to be given here. A summary would help.   

Line 81 - these drugs. I assume you are talking about chloramphenicol, tetracycline etc. If so, please reiterate at this point so that people are clear as to which drugs you are talking about.  

Lines 125-137 are very hard to read. I think tabulating them would really help in getting the message across. A bar graph would be even better. I think the message about chloramphenicol being less effective should come at the top of the parpagraph.   

Line 192 - should you call it fungal load when P. insidiosum is not really a fungus? 

Line 202 - what immunotherapy? Please name it. 

Table 2 - Study at LV Prasad - how many cases? 

Table 2 - study of 46 cases (ref 53). How many got each of the 4 treatments a-d

Tabel 2 (ref 54) - name the immunotherapy 

Table 2- (ref 56) - need number of cases and numbers in addition to the percentages in the last column (outcomes and notabel points)

Also make sure that in Table 2 - taht all the studies are separate studies. That the cases aren't in 2 studies. Any overlapping studies should be removed as it is unclear as to what belongs where and the outcomes are unclear. need stand-alone clean studies so the true efficacy can be determined. 

Ref 66 (Table 2). Need to show the survival in the 2 arms so we can see if it is significnat or not.   

Line 45 - the word is highlighting not spotlighting 

Author Response

Dear Reviewer

We acknowledge the receipt of your review and thank you for the time spent evaluating our manuscript. Your comments are noted and appreciated for the critical insights they offer.

The responses to your observations have been compiled and can be found below. These points have been addressed and marked within the manuscript accordingly. The feedback has been instrumental in guiding the revisions, and we have tried to incorporate the necessary changes.

Once again, thank you for your contribution towards improving our work.

Q1. Major comments

This is a nice review. However, it needs some refinement to improve the paper.

 The immunotherapies (lines 42-44) need to be expandded. Details of what they are and their effect need to be given here. A summary would help.

Thank you for your comments regarding including more details about immunotherapy in our manuscript. We understand the importance of providing a comprehensive overview of all relevant treatment modalities for pythiosis, including immunotherapy. However, it's important to note that the primary focus of our work is not on the detailed description of immunotherapeutic approaches. There are already well-established reviews on this topic (doi 10.5411/wji.v4.i2.88 and doi 10.3390/vaccines9101080).

Despite this, to provide a better understanding of pythiosis treatments for readers who may not be familiar with this field, we have added a new section called "Overview of Pythiosis Treatment". This section includes relevant information about immunotherapy and provides a general perspective on its role and effectiveness. Additionally, we have included other treatment modalities for pythiosis to give readers a more comprehensive understanding of the therapeutic landscape for this infection. This addition aims to balance the depth of information with the broader scope of our study's objectives.

Line 81 - these drugs. I assume you are talking about chloramphenicol, tetracycline etc. If so, please reiterate at this point so that people are clear as to which drugs you are talking about. 

The text was rewritten for clarity.

 Lines 125-137 are very hard to read. I think tabulating them would really help in getting the message across. A bar graph would be even better. I think the message about chloramphenicol being less effective should come at the top of the parpagraph.  

Agreed. The text has been revised, and a graphic has been created to illustrate the data more effectively.

 Line 192 - should you call it fungal load when P. insidiosum is not really a fungus?

Replaced (microbial load).

 Line 202 - what immunotherapy? Please name it.

                Named (Pitium-Vac® immunotherapy)

 Table 2 - Study at LV Prasad - how many cases?

                Added (114 patients)

 Table 2 - study of 46 cases (ref 53). How many got each of the 4 treatments a-d

                The information was updated.

 Tabel 2 (ref 54) - name the immunotherapy

                The information was updated.

 Table 2- (ref 56) - need number of cases and numbers in addition to the percentages in the last column (outcomes and notabel points)

                The text was rewritten.

 Also make sure that in Table 2 - that all the studies are separate studies. That the cases aren't in 2 studies. Any overlapping studies should be removed as it is unclear as to what belongs where and the outcomes are unclear. need stand-alone clean studies so the true efficacy can be determined.

Dear Reviewer,

We understand and agree that overlapping data could potentially obscure the interpretations and conclusions regarding the efficacy of the interventions studied. However, we would like to clarify that our manuscript's objective does not encompass the reassessment of data duplication in previously published articles, as these articles have undergone their own peer review processes where such issues should have been addressed. In our current manuscript, to the best of our knowledge, have not included identical studies in Table 2.

                Nonetheless, we appreciate your concern and, If you could kindly specify which studies you suspect might overlap or require further clarification, we would be more than willing to review our dataset and make the necessary adjustments to ensure the integrity and clarity of our findings.

 Ref 66 (Table 2). Need to show the survival in the 2 arms so we can see if it is significnat or not. 

                The text was update.

 Q2. Detail comments

Line 45 - the word is highlighting not spotlighting

Replaced.

Round 2

Reviewer 2 Report

The changes enhance the paper. They address my concerns.  

Nil else